# A Comparative Transcriptomic Meta-Analysis Revealed Conserved Key Genes and Regulatory Networks Involved in Drought Tolerance in Cereal Crops

**DOI:** 10.3390/ijms222313062

**Published:** 2021-12-02

**Authors:** Elena Baldoni, Giovanna Frugis, Federico Martinelli, Jubina Benny, Donatella Paffetti, Matteo Buti

**Affiliations:** 1National Research Council (CNR), Institute of Agricultural Biology and Biotechnology (IBBA), Via Alfonso Corti 12, 20133 Milan, Italy; 2National Research Council (CNR), Institute of Agricultural Biology and Biotechnology (IBBA), Rome Unit, Via Salaria Km. 29,300, 00015 Monterotondo, Italy; giovanna.frugis@cnr.it; 3Department of Biology, University of Florence, 50019 Sesto Fiorentino, Italy; federico.martinelli@unifi.it; 4Department of Agricultural, Food and Forest Sciences, University of Palermo, 90133 Palermo, Italy; jubina.benny@unipa.it; 5Department of Agriculture, Food, Environment and Forestry (DAGRI), University of Florence, 50144 Florence, Italy; donatella.paffetti@unifi.it

**Keywords:** rice, barley, *Brachypodium*, maize, drought tolerance, comparative transcriptomics, *stay-green rice*, photosynthesis, transcription factors

## Abstract

Drought affects plant growth and development, causing severe yield losses, especially in cereal crops. The identification of genes involved in drought tolerance is crucial for the development of drought-tolerant crops. The aim of this study was to identify genes that are conserved key players for conferring drought tolerance in cereals. By comparing the transcriptomic changes between tolerant and susceptible genotypes in four Gramineae species, we identified 69 conserved drought tolerant-related (CDT) genes that are potentially involved in the drought tolerance of all of the analysed species. The CDT genes are principally involved in stress response, photosynthesis, chlorophyll biogenesis, secondary metabolism, jasmonic acid signalling, and cellular transport. Twenty CDT genes are not yet characterized and can be novel candidates for drought tolerance. The k-means clustering analysis of expression data highlighted the prominent roles of photosynthesis and leaf senescence-related mechanisms in differentiating the drought response between tolerant and sensitive genotypes. In addition, we identified specific transcription factors that could regulate the expression of photosynthesis and leaf senescence-related genes. Our analysis suggests that the balance between the induction of leaf senescence and maintenance of photosynthesis during drought plays a major role in tolerance. Fine-tuning of CDT gene expression modulation by specific transcription factors can be the key to improving drought tolerance in cereals.

## 1. Introduction

Cereal crops are a major source of food for humans and livestock throughout the world (FAO statistics, http://www.fao.org/faostat/en/#data, accessed on 30 September 2021). Due to the increasing world population, enhancing food security is an urgent need; however, the production of cereals is endangered by global warming and related abiotic stressors, such as heat, drought, and salinity [1,2]. Drought is one of the most critical global warming-related stress factors affecting crop growth and development, and leads to severe yield loss [3,4]. Water deficiency is a global concern since even the most productive agricultural regions can occasionally experience short periods or years of severe drought. Furthermore, irrigation use might be restricted in the future due to the competition from non-agricultural activities, such as industry and urbanisation [5]. In the future, the potential impact of climate change on rainfall patterns and the need to extend the exploitation of marginal lands will further reduce the availability of water for agriculture. Therefore, a large effort is required to increase agricultural production by improving the adaptation of crops to sub-optimal environments in spite of reduced water availability [4,5].

Plants have evolved several morphological, physiological, biochemical, and molecular mechanisms to overcome drought stress. At a physiological level, drought stress prevention mechanisms are aimed at balancing water absorption and loss. The two prevention mechanisms that favour the absorption of water are enhanced root growth and the accumulation of solutes to reduce the water potential in the tissues. Plants accumulate different types of organic and inorganic solutes in the cytosol, such as proline, sucrose, and glycine betaine, to lower cellular osmotic potential, thereby maintaining cell turgor and improving water uptake from the drying soil [6,7]. In particular, proline acts not only as an excellent osmolyte but also as a metal chelator, an antioxidant, and a signalling molecule [8]. On the other hand, mechanisms enhancing stomatal closure, the reduction of shoot growth, and the acceleration of leaf senescence are triggered to limit the water loss [9]. These mechanisms reduce transpiration rates, which negatively influences the absorption of CO_2_ [10]. A common effect of water stress is the alteration of the balance between the generation and quenching of reactive oxygen species (ROS) [11]. Increased ROS levels can cause lipid peroxidation, protein denaturation, DNA mutations, and various types of cellular oxidative damage. During water stress, both non-enzymatic (e.g., ascorbate and glutathione) and enzymatic (e.g., superoxide dismutase and catalase) antioxidant systems are activated in plant species [11]. In addition, flavonoids and hydroxycinnamates, two important classes of phenylpropanoids, are produced by plants during stress for their role in ROS scavenging [11]. 

Water stress significantly affects intra-cellular hormone contents [12,13]. Abscisic acid (ABA) plays a prominent role in the response to water stress, mediating the crosstalk among phytohormones and other signalling pathways [14]. Although ABA is the most characterized drought-responsive hormone, the role of jasmonate and ethylene in the water stress response is emerging [13]. The crosstalk between the different plant hormones results in synergetic or antagonistic interactions that play crucial roles in the water stress response [14]. 

At a molecular level, the expression of many genes is modulated by water stress conditions, and recent studies elucidated the mechanisms of drought response in plants [15,16]. These genes encode for proteins with a regulatory/modulation function on gene expression (e.g., transcription factors (TFs), protein kinases, and other proteins involved in signal transduction), and proteins and enzymes with a direct role in protecting cell structures, such as (i) late embryogenesis abundant (LEA) proteins and chaperones, which stabilize cellular components during abiotic stress; (ii) detoxifying enzymes involved in the protection from oxidative damage (e.g., superoxide dismutase, catalase, and glutathione peroxidase); (iii) enzymes for the synthesis of osmoprotectants (e.g., proline, glycine-betaine, soluble sugars, and trehalose). The accumulation of these functional metabolites activates tolerance mechanisms like ROS-scavenging and osmotic adjustment in the whole plant [8,11]. ABA is a master regulator of this complex gene regulatory network. Two principal stress response mechanisms, the ABA-dependent and the ABA-independent signalling pathways, were first discovered in *Arabidopsis thaliana* [15] and characterized in crops, including rice [16,17]. In the ABA-dependent pathway, the ABA-responsive element (ABRE) is the major cis-element for ABA-responsive gene expression. ABRE-binding factors (ABFs) are bZIP-type TFs that control gene expression in an ABA-dependent manner [16]. On the other hand, analysis of the promoter regions of genes showing ABA-independent expression in stress responses and tolerance has shown the presence of the DRE cis-element. AP2/ERF TFs were identified as DRE-binding factors (DREB), which specifically interact with the DRE sequence and control the expression of a large number of stress-responsive genes [16]. In addition to the DREB and ABF regulons, the NAM-ATAF1/2-CUC2 (NAC) regulon plays a major role in the transcriptional network of the dehydration response, and stress-responsive NAC (SNAC) TFs are involved in abiotic stress tolerance [18]. Other TF families have been extensively reported to be involved in drought response, including the APETALA2/Ethylene-responsive element-binding protein (AP2/EREBP) [19], basic leucine zipper (bZIP) [20], MYB [21], WRKY [22], and zinc finger [23] TFs. Research advances have elucidated the role of molecular/cellular signalling networks in drought response due to a strong interconnection of signals from hub TFs, MAPK pathways, ROS, and lipid-derived pathways [24].

While knowledge about the complex mechanisms regulating drought tolerance in plants has significantly improved lately, the development of water-saving and drought-tolerant cereal crops to cope with water shortage and impending demand for food production still remains extremely challenging. This is mostly due to the fact that drought tolerance is a complex trait controlled by a large group of genes, and is strongly influenced by genotype and environmental interactions [25,26]. In recent years, transcriptomic studies comparing genotypes with contrasting drought-response phenotypes (sensitive vs. tolerant) have been performed in many cereal species [27,28,29,30,31,32,33]. The use of contrasting genotypes allows for discriminating the genes and pathways that are effective for drought tolerance. Comparative transcriptomics allows for the exploitation of the natural variation present in genotypes worldwide, also analysing the ecotypes better adapted to sub-optimal environments. The genes that are differentially regulated between the drought-tolerant and the drought-sensitive genotypes can be putative candidates for further characterizations and allele mining, and to discover novel alleles for breeding programs. The huge availability of transcriptomic data promotes the use of comparative meta-analyses, allowing for the investigation of the effects of the same environmental cue across different studies. This approach is very useful for dissecting metabolic and regulatory pathways related to stress tolerance [34,35,36,37].

Here, we performed a comparative transcriptomic meta-analysis related to four Gramineae species, i.e., Brachypodium (*Brachypodium distachyon* L.), barley (*Hordeum vulgare* L.), rice (*Oryza sativa* L.), and maize (*Zea mays* L.), using RNA-Seq data available in public repositories. The aim of this study was to identify genes that are key players for improving drought tolerance as well as upstream TFs involved in the regulation of these genes.

## 2. Results

### 2.1. Selection of RNA-Seq Datasets for Meta-Analysis

To identify the genes that may represent key players for conferring drought tolerance in different Gramineae species, a transcriptomic meta-analysis was performed starting with published datasets with comparable characteristics. Four datasets that were developed from four different species were selected—Brachypodium (*B. distachyon*) [30], barley (*H. vulgare*) [29], maize (*Z. mays*) [31], and rice (*O. sativa*) [32]. The four experiments aimed to characterize the transcriptomic responses of two contrasting genotypes (one tolerant and one sensitive) to drought stress. In these experiments similar growth conditions and stress treatments were adopted, as described in detail in Section 4.1, and the total RNA was extracted from the leaves of the stressed and non-stressed plants of sensitive and tolerant accessions at the vegetative stage, and subsequently sequenced. Table 1 summarizes the main features of the materials used for this study.

The overall workflow from data mining to the identification of candidate genes is displayed in Figure 1. 

### 2.2. RNA-Seq Data Processing and Differential Expression Analyses

Briefly, after assessing the high quality of the RNA reads, adapters and low-quality reads were filtered out, resulting in a percentage of “survived” reads of about 88% for rice, 82–91% for maize, 95–97% for barley, and 99% for Brachypodium datasets (Appendix A). For each RNA library, 84–97% of filtered reads aligned to the respective reference genome assembly (Appendix A), and the number of reads mapping to each predicted gene was estimated.

Differential expression analyses of stressed vs. control samples for both tolerant and sensitive accessions were carried out (Appendix A). The number of up- and down-regulated differentially expressed genes (DEGs) for each analysis, along with the number of active genes, is summarised in Table 2. 

In rice, the two contrasting genotypes displayed a similar number of DEGs. In contrast, the tolerant genotypes of Brachypodium and maize showed a lower number of DEGs compared to the sensitive ones, whereas in barley, the tolerant genotype showed a higher number of DEGs than the sensitive one.

The DEGs of each species were then mapped to the rice predicted genes to obtain comparable datasets. For each genotype, the number and percentage of DEGs orthologous to the rice genes are shown in Appendix A.

### 2.3. Comparison of the Level of Drought Stress among the Four Datasets

We used the four datasets derived from the rice orthology analysis to evaluate the comparability of the drought treatments imposed on the plants in the considered experiments. First, we compared the DEGs among the four experiments, considering all the genes that were differentially expressed in at least one genotype of each experiment. A total of 142 DEGs common among the four experiments was found (Appendix A). Then, we referred to the paper by Todaka et al. [38] to evaluate the drought severity of the four treatments. The authors analysed the physiology, gene expression, and metabolic changes in rice plants at the vegetative stage subjected to drought treatments with different intensities. In particular, the moderate level 3 treatment (called Md3 in [38]) caused stomatal closure and shoot growth retardation, while the severe drought treatment (called Sds in [38]) also resulted in leaves wilting and photosynthesis inhibition. We compared the rice DEGs under the Sds treatments from Todaka et al. with the DEGs of the four selected datasets, considering the orthology of the rice genome for Brachypodium, barley, and maize. A total of 1377, 407, 1242, and 1140 DEGs resulted between the Sds treatment [38] and the considered experiments of Brachypodium, barley, maize, and rice, respectively. In addition, 65 out of the 142 common DEGs among the four experiments (Appendix A) were differentially expressed under the Sds treatment [38]. Therefore, the transcriptomic response of the plants analysed in the four selected experiments was consistent with a severe condition of drought stress.

### 2.4. Identification of Genes Involved in Tolerance to Drought Stress in each Species

We assumed that the genes putatively involved in the tolerant response to drought are the ones whose expression regulation differs between sensitive and tolerant accessions during the drought treatment, as previously reported by Buti et al. [35]. Then, we classified the DEGs putatively involved in drought tolerance in eight classes, according to the log_2_(Fold Change) (LFC) difference between the sensitive and tolerant genotypes of the DEGs, as follows: (i) genes up-regulated in the sensitive genotype and unchanged in the tolerant one (“sen+”); (ii) genes down-regulated in the sensitive genotype and unchanged in the tolerant one (“sen−”); (iii) genes up-regulated in the tolerant genotype and unchanged in the sensitive one (“tol+”); (iv) genes down-regulated in the tolerant genotype and unchanged in the sensitive one (“tol−”); (v) genes differentially expressed in both genotypes, showing a difference of LFC (LFC of tolerant sample—LFC of sensitive sample) higher than 1 (“∆LFC > 1”); (vi) genes differentially expressed in both genotypes, showing a difference of LFC lower than -1 (“∆LFC < −1”); (vii) genes down-regulated in the sensitive genotype and up-regulated in the tolerant one (“sen−/tol+”); (viii) genes up-regulated in the sensitive genotype and down-regulated in the tolerant one (“sen+/tol−”). The complete classification is reported on Appendix A. The numbers of genes belonging to the eight classes are summarized in Table 3. 

Among these genes, 5337 Brachypodium genes (79.15%), 1557 barley genes (71%), and 4723 maize genes (84.22%) displayed a putative ortholog in the rice genome (Appendix A).

### 2.5. Identification of Genes Involved in Tolerance to Drought Stress across Multiple Species

To identify conserved genes involved in drought tolerance across multiple species, the IDs of the rice orthologous genes, previously identified and reported in Appendix A, were compared among the four datasets. We identified 69 genes that were present in all of the four species (Appendix A) (hereafter referred to as CDT genes). Considering rice as the most characterized species, several CDT genes are known to be involved in specific cell functions (Table 4). 

As expected, some CDT genes are known to be involved in the abiotic stress response, including the ABA-responsive gene *RD22* (Os01g0733500) [39], and two heat shock protein-encoding genes (HSPs), *HSP70* (Os02g0710900 [40]), and *OsHsp60-1* (Os10g0462900). Several genes play a role in different photosynthesis-related processes. Three genes are involved in the assembly of photosystems. Os03g0592500 encodes a light-harvesting chlorophyll-binding protein (LHCB) of the photosystem II; Os04g0414700 and Os07g0148900 encode the photosystem I subunits O (PsaO) and K (PsaK) [41], respectively. The gene Os01g0639900 codes for a carbonic anhydrase 1 (*OsCA1*), involved in CO_2_ fixation in RuBP through Rubisco activity [42]. Some genes are involved in the Calvin cycle. Os04g0459500 and Os03g0129300 code for glyceraldehyde-3-phosphate dehydrogenases (G3PDH), Os04g0234600 codes for a sedoheptulose 1,7-bisphosphatase (OsSBP), and Os05g0186300 codes for the NADP-dependent malic enzyme OsNADP-ME3. This enzyme catalyses the oxidative decarboxylation of malate and NADP^+^ in the chloroplast, releasing CO_2_, which is then utilized in the Calvin cycle [43]. Other genes are involved in leaf senescence, including the *stay-green rice* gene (*OsSGR*, Os09g0532000), a chlorophyll-degrading Mg^++^-dechelatase [44,45,46]. The gene *OsHemA1* (Os10g0502400), a glutamyl-tRNA reductase involved in the initial steps of chlorophyll synthesis [47], and the gene *OsGDCH* (glycine decarboxylase complex H subunit, Os10g0516100) is involved in leaf senescence [48]. Two gene codes for chloroplast proteins that play important roles in oxidative stress protection are *OsFd1* (Os08g0104600), a ferredoxin that mediates electron transfer and probably regulates the expression of plastidial genes in a redox-dependant manner [49], and *CSP41b* (Os12g0420200), an NAD^+^(P)-binding protein which is required for normal leaf colour and chloroplast morphology [50]. In addition, *OsNOX7* (Os09g0438000) codes for an NADPH oxidase-respiratory burst oxidase homologue (RBOH) protein [51]. Three genes are involved in secondary metabolism, including the upstream *PAL2-2* gene (Os02g0626400) [52], the S-adenosylhomocysteine hydrolase gene (*OsSAHH*, Os11g0455500), which is involved in lignin biosynthesis [53], and the chalcone-flavonone isomerase (CHI, Os12g0115700), which is involved in flavonoid biosynthesis [54]. Three genes are involved in hormone signalling. Os08g0508800 and Os03g0700700 code for two lipoxygenases (*OsHI-LOX* and *OsLOX8*, respectively), are involved in jasmonic acid (JA) biosynthesis, and are important players in the JA signalling pathway [55,56], while Os11g0145200, coding for an indole-3-acetate beta-glucosyltransferase, is putatively involved in auxin signalling [57]. Five genes (Os02g0465900, Os03g0114800, Os04g0390100, Os06g0129400, and Os06g0486800) are probably involved in cellular transport. Finally, twenty CDT genes have not been characterized yet (Table 4).

### 2.6. Principal Component Analysis and Protein–Protein Interaction of CDT Genes

To investigate whether the expression profiles of the CDT genes could intercept the differences between tolerant and sensitive genotypes, we performed a principal component analysis (PCA) for each dataset using the expression data of the CDT genes (Figure 2).

In Brachypodium (Figure 2A), the first PCA component (Dim1) accounted for 67.3% of the variance, clearly separating the stressed sensitive samples from the other ones. The second component (Dim2) captured a variance of 11%, differentiating the control sample of the sensitive genotype from the other ones. Gene expression of the CDT genes was very similar in stressed and control samples of the *B. distachyon* tolerant genotype. In barley (Figure 2B), Dim 1 (39.3%) separated stressed samples from the control ones independently from the genotype, whereas Dim2 (26.5%) separated the stressed samples of the tolerant genotype from all the others. A similar situation was observed in rice (Figure 2C), where the first component (54.7%) clearly separated the stressed from the control samples of both genotypes, whereas Dim2 (27.6%) intercepted the differences in control samples between tolerant and susceptible genotypes. In maize (Figure 2D), Dim1 (50.9% of the variance) separated the samples of the tolerant genotype from those of the sensitive genotype. Dim2 (32.4% of the variance) clearly separated the control samples from the stressed ones in the susceptible genotypes, while not clearly separating the stressed and control samples of the tolerant genotype. 

A protein–protein interaction (PPI) network using the 69 rice CTD genes revealed that 23 proteins showed at least one interaction (Figure 3). 

A principal network of 12 proteins that interacted with each other was identified. This main network included the three photosystem-related proteins PsaK (OsJ_23098), PsaO (OsJ_14748), and LHCB (RCABP89), which were strongly interconnected with each other, the two G3PDHs (OsJ_15048 and OsJ_09272), OsFd1 (ADI1), CSP41b (OsJ_35887), OsGDCH (GDCSH), and the protein Os03g0844900 (OS03T0844900-01). Moreover, the two LOX proteins (OsJ_12241, CM-LOX1) and the two HSPs (OS02T0710900-01, OsJ_31804) constituted two independent networks.

### 2.7. Clustering Gene Expression to Find Common Patterns and Associated Transcription Factors

To identify putative upstream regulators of the differentiation between tolerant and sensitive responses, we selected the TF-encoding DEGs of the four species (Appendix A) and we performed a k-means cluster analysis of the gene expression profiles of the CDT genes together with the TF–DEGs for each species. The optimum cluster number was determined based on converging results of the sum of squared errors (SSE) estimate and the Calinsky criterion, as described by Testone et al. [58]. The analysis defined the presence of four different expression clusters for each species (Figure 4, Appendix A). 

Differences in gene expression behaviour between sensitive and tolerant genotypes were highlighted by the cluster profiles. In addition, some clusters were strongly negatively correlated, including cluster 1 and cluster 2 in Brachypodium (r = −0.98) (Figure 4A), cluster 1 and 2 in barley (r = −0.94) (Figure 4B), cluster 1 and 3 in rice (r = −0.9) (Figure 4C), and cluster 2 and 4 in maize (r = −0.98) (Figure 4D). The CDT genes showed a peculiar cluster distribution in each species, but some similarities were detected; in Brachypodium, barley, and rice, the photosynthetic genes *PsaO*, *PsaK*, *LHCB*, and *CA1* clustered together (green arrows in Figure 4). Similarly, in maize, the genes *PsaO*, *LHCB*, and *CA1* belonged to cluster 4, whereas *PsaK* clustered separately in cluster 3. Interestingly, in each species, the cluster harbouring the photosynthesis-related genes (cluster 2 in Brachypodium, barley, and rice, and cluster 4 in maize) was strongly negatively correlated with the cluster containing the homologue gene of rice *SGR* (brown arrows in Figure 4). 

We then focused on the TF-encoding genes belonging to the two negatively correlated clusters related to photosynthesis and *SGR* to identify possible TFs with a regulatory role in the drought-tolerant response. In the selected clusters, we detected 265, 51, 84, and 142 TF-encoding genes for Brachypodium, barley, rice, and maize, respectively (Appendix A). Most of these genes belonged to the eight classes previously defined (264, 42, 70, and 141 for Brachypodium, barley, rice, and maize, respectively). The distribution of the principal TF families in the two negatively correlated species of the four species is shown in Appendix A. The analysis highlighted that several TF families (AP2, ARF, bHLH, bZIP, C3H, DBB, ERF, G2-like, HD-ZIP, HSF, MYB, and NAC) were represented in the two negatively correlated clusters of all four species, and genes belonging to the bHLH, bZIP, MYB, and NAC families were the most abundant. 

We used a not stringent orthology analysis (see Section 4.4 for details) to identify the similar TF-encoding genes that may share conserved functions related to drought tolerance across the four species. We detected 18 groups of DEGs coding for TFs, five of which were represented in three of the four analysed species (Table 5).

### 2.8. Gene Co-Expression Network Analysis of CDT Genes and TF DEGs on an Independent Water Stress Experiment in Rice 

We focused on rice to validate these results with independent data and to find regulatory relationships between the identified CDT genes and specific families of TFs. Hence, we constructed and analysed a targeted gene co-expression network (GCN) related to the transcriptomic data of a previous experiment of PEG-mediated osmotic stress in two rice genotypes, one tolerant (Eurosis) and one sensitive (Loto) to osmotic stress [28]. In particular, the genes encoding TFs that were differentially expressed under osmotic stress in both genotypes and the 69 CDT genes were selected, for a total of 797 genes. The list of the TF-encoding genes used for the analysis and their gene expression values, together with those of the rice CDT genes, are reported in Appendix A. Pairwise correlation analysis of their expression values was performed (Appendix A), and significant correlations (*p*-value < 0.05) and absolute Pearson’s correlation (|r|) values ≥ 0.9 were used to develop the GCN. The Cytoscape platform [59] was used to determine the relationships among the selected genes and to identify putative upstream regulators of the genes related to photosynthesis and senescence that differentiate the tolerant from the sensitive responses. The data related to the network topological analysis are shown in Appendix A. The GCN network included 787 out of the initial 797 genes (Figure 5A). 

In the GCN, the *PsaO* gene had the highest number of connections (429) and formed a core subnetwork of two negatively correlated groups of genes, including 43 of the 69 CDT genes (Appendix A). Among them, 26 CDT genes were positively correlated with *PsaO* and included several genes involved in photosynthesis—*PsaK*, *LHCB*, *OsCA1*, and three genes involved in the Calvin cycle (the *OsSBP* gene Os04g0234600 and the two *G3PDH* genes, Os04g0459500 and Os03g0129300) found in the PPI network. Other 16 CDT genes, including SGR, were negatively correlated to *PsaO*. These genes were positively correlated to SGR (Appendix A), fully corroborating the k-means clustering data. Node degree, which, in a GCN network, is the number of neighbours to which a node directly connects, is an important centrality parameter that identifies essential genes in the network [58,60]. In this respect, the GCN analysis identified *PsaO, PsaK*, and *LHCB*, with node degrees of 429, 418, and 410 respectively, as hub central genes in the osmotic stress response. 

Among the 84 rice TF-encoding genes belonging to the two negatively correlated clusters related to photosynthesis and *SGR* in the k-means cluster analysis (rice clusters 2 and 3, respectively; Appendix A), 68 genes were included in the GCN. In particular, 14 TF genes in the GCN were strongly positively and negatively correlated (|r| values ≥ 0.9) to *PsaO* and *SGR*, respectively (Appendix A). Amongst them, genes belonging to the nuclear factor Y (NF-Y or NF-YB/NF-YC) group displayed high node degrees in the GCN and were highly co-expressed with the *PsaO*, *PsaK*, and *LHCB* genes. A member of the *CONSTANS-like* (*COL*) TF gene family (Os03g0711100) was highly co-expressed with three NF-Y genes in the *PsaO* core subnetwork. Os03g0711100 showed a high betweenness centrality (BC) score in the global GCN network, which indicates a key role in the transcriptional regulatory network. On the other hand, 20 TF genes strongly positively and negatively correlated (|r| values ≥ 0.9) to *SGR* and *PsaO*, respectively (Appendix A). Amongst them, genes belonging to the NAC, MYB, HSF, ERF, and bZIP families also displayed high node degrees in the GCN and were highly co-expressed with *SGR*. 

### 2.9. Analysis of TF Binding Sites on the Promoter Sequences of the Rice CDT Genes 

The 3000 bp upstream nucleotide sequences of the rice CDT genes (Appendix A) were searched for the binding sites of the NF-Y, NF-CO complexes and the NAC, MYB, ERF, and bZIP TF families (Appendix A). Several NF-Y and NF-CO motifs were present in the promoters of the CDT genes belonging to the photosynthesis-related cluster. The promoters of *PsaO* and *PsaK* were particularly enriched in NF-CO regulatory elements, especially within the 500 bp proximal promoter region. The promoters of *LHCB*, *Os_25*, *Os_26*, *Os_29*, *Os_34*, *Os_46*, and *Os_60* were also enriched in the NF-Y binding sites and may be co-regulated with *PsaO* and *PsaK* as part of the same NF-Y and NF-CO regulatory module. The promoter of *PsaO* was also enriched in cis-regulatory elements for bZIP, ERF, and MYB-related classes of TFs. The *SGR* promoter was characterized by several ABRE motifs and numerous consensus sequences for the NAP TF binding site (TACGT) and for MYB TFs, in particular, to those involved in dehydration and water stress response, like Arabidopsis MYB DOMAIN PROTEIN 2 (AtMYB2) (motifs MYB1AT and MYB2AT in Appendix A) [61]. 

## 3. Discussion

Drought is a major environmental constraint for worldwide agriculture, and cereals are subjected to high yield loss due to this abiotic stress. As the world population is expected to reach nine billion people by 2050, crop yields need to be improved by 40% in areas where drought is likely to occur by 2025 [2,4]. The development of crops with increased drought tolerance can be achieved through advanced molecular breeding techniques or biotechnological approaches [17,62]. The prerequisite for the application of these strategies is the identification of drought-tolerant genes and the dissection of the biological mechanisms in which these genes are involved. Several efforts have been made in this direction, and one recent approach consists of meta-analyses of large transcriptomic datasets [34,35,36,37]. These analyses usually compare many experiments related to one species using data obtained with different cultivation methodologies, drought treatments, plant stages, analysed tissues, modalities, and timings of sampling, which often precludes the identification of useful targets for breeding programs. Instead, we selected four datasets related to four different cereal species that considered similar cultivation methodologies (i.e., plants grown in soil), drought treatments (i.e., severe stress for several days), plant stages (i.e., vegetative stage), and analysed tissues (i.e., leaf), to focus on specific drought tolerance mechanisms that are conserved among cereal species [29,30,31,32]. In addition, the selected transcriptomic studies were all performed on two contrasting genotypes since the comparison of genotypes showing contrasting responses to stress (tolerance and sensitivity) is crucial for identifying key players in the tolerant response [27,28,33,63]. 

DEG analyses of the four datasets gave results comparable with the original ones [29,30,31,32]. The differences observed in the DEG numbers among the four species may be due to the different durations of the four drought treatments, since the amplitude of the activated transcriptome reflects the phases of the stress response [64]. Indeed, at the beginning of the stress treatment, sensitive genotypes are more conditioned by drought than tolerant ones and can undergo higher transcriptional fluctuations. After prolonged drought, the tolerant genotypes also sensed the stress with a consequent increase of gene expression fluctuations [64]. Consistently, the species subjected to a shorter treatment (8 and 7 days for Brachypodium and maize, respectively) showed fewer DEGs in tolerant genotypes, whereas in the barley experiment, where the stress experiment was more prolonged (30 days), the tolerant genotype showed a higher number of DEGs than the sensitive one. Nevertheless, one-to-one orthology analysis with the rice genome and the comparison with the drought treatments described by Todaka et al. [38] showed that the four experiments were comparable in terms of stress severity.

### 3.1. CDT Genes Characterized the Drought Response of Sensitive and Tolerant Genotypes in the Four Cereal Species 

Based on the assumption that the expression of genes involved in the tolerant response to drought differs between sensitive and tolerant genotypes during the occurrence of the stress [35], we classified the DEGs putatively involved in drought tolerance in eight classes, according to their expression regulation under drought (Table 3). When the classified DEGs were compared among species, 69 genes that are possibly involved in the differentiation between drought-sensitive and tolerant responses and conserved among the four species were detected and called conserved drought tolerant-related (CDT) genes (Appendix A). Thirty-six CDT genes were differentially expressed in the leaves of rice plants subjected to the Sds treatment reported in [38] (Appendix A), confirming their relevant role in response to severe drought stress. PCA (Figure 2) and cluster analysis (Figure 4) of the CDT gene expression confirmed the ability of these genes to differentiate the tolerant from the sensitive genotypes in the four species by highlighting possible differences in the mechanisms of tolerance among the four species. Tolerant Brachypodium and maize genotypes displayed weak or no fluctuation of CDT gene expression in contrast to the dramatic changes in the sensitive genotypes, whereas, in barley and rice, both sensitive and tolerant genotypes displayed gene expression changes in response to the stress treatment. 

Several CDT genes are known for their role in different biological processes, including photosynthesis, response to oxidative stress, chlorophyll biosynthesis and degradation, biosynthesis of secondary metabolites, hormone signalling, and cellular transport (more details are provided in Section 2.5). Several CDT genes involved in photosynthesis were strongly interconnected. In the principal PPI network (Figure 3), PsaO, PsaK, LHCB, the G3PDH Os04g0459500, and the SBP showed a high number of interactions, suggesting an important role within the network. The associations among *PsaO*, *PsaK*, and *LHCB* were confirmed by the k-means cluster analysis of gene expression (Figure 4 and Appendix A). These three genes, together with *CA1*, which is involved in the CO_2_ fixation of photosynthesis, belonged to the same clusters in the four species (clusters 2, 3, 4, and 2 for Brachypodium, barley, maize, and rice, respectively), with the exception of *PsaK* in maize, which clustered independently in cluster 4. This observation could be due to differences in the gene regulatory networks related to photosynthesis, since maize is a C4 species, while Brachypodium, barley. and rice are C3 [65]. These groups of genes were negatively correlated to *SGR* homologous genes in the four species (i.e., belonging to the negatively correlated clusters). *SGR* is a major component of the chlorophyll degradation pathway and represents a key regulator of the leaf senescence process [44,45]. These data suggest the importance of photosynthesis and leaf senescence-related mechanisms in the differentiation between sensitive and tolerant responses to drought in cereals.

Among the CDT genes, 20 are not yet characterized (Table 5). Some of them are reported as chloroplastic proteins, and the putative orthologs of Arabidopsis are related to the assembly and functioning of photosystem II [66]. Most of these genes were positively correlated with the four photosynthetic genes *CA1*, *PsaO*, *PsaK*, and *LHCB*, and negatively correlated with *SGR* (Appendix A), suggesting a role in photosynthesis. In addition, one of these uncharacterized genes, Os05g0468900, was previously detected among genes that are involved in the response to different abiotic stresses of contrasting rice genotypes [35], confirming an important role of this gene in the differentiation between sensitive and tolerant responses to environmental cues. 

### 3.2. The Balance between the Induction of Leaf Senescence and the Maintenance of Photosynthesis Plays a Major Role in Drought Tolerance

Our findings point to a major role of the balance between *SGR*-mediated leaf senescence and *PsaO*/*PsaK*/*LHCB*/*CA1*-mediated photosynthetic function in drought stress tolerance. Physiological leaf senescence is the final phase of the development of a leaf, where a self-destructive program to degenerate the cellular structures is activated and allows a leaf to make its final contribution to the plant by remobilizing the nutrients accumulated in the senescing leaf. During leaf senescence, several morphological, physiological, and molecular changes occur; photosynthesis is down-regulated, whereas nitrogen remobilization is up-regulated. The expressions of a high number of genes, called senescence-associated genes (SAGs), are modulated [67]. The four photosynthetic genes *PsaO*, *PsaK*, *LHCB* and *CA1* are reported as SAGs, as expected because of the down-regulation of photosynthetic activity [68]. Several other CDT genes were described as SAG [68] (Appendix A). These genes are involved in the biological processes related to leaf senescence, such as chlorophyll biosynthesis, oxidative stress response, photosynthesis, JA-mediated signalling and cellular transport [69]. The regulation of leaf senescence has great importance in the life cycle of the plants and in terms of crop productivity since premature leaf senescence negatively impacts the yield stability, whereas delayed senescence and longer maintenance of photosynthesis can improve grain yield. Leaf senescence can be activated by abiotic stresses, including drought, through an ABA-mediated mechanism [70,71]. It has been demonstrated that the suppression of drought-induced leaf senescence in annual plants results in improved drought tolerance and minimal yield losses [72,73]. Hence, the balance between the induction of leaf senescence and the maintenance of photosynthesis can play a major role in drought tolerance and in preserving crop yields during stress.

### 3.3. Identification of Transcription Factors Involved in the Balance between Leaf Senescence and Photosynthesis 

TFs are key players in the regulatory networks underlying plant responses to abiotic stresses [21,24]. Moreover, the importance of transcription regulation in the onset of leaf senescence is well-established [74]. TF-encoding genes were not found among the identified CDT genes, likely because for multigene families, such as TF families, the definition of orthologous relationships is a challenging task, and the commonly used predictive algorithms may fail the identification. However, the expressions of several TF genes were found as strictly associated with the specific CDT genes of the four species in the cluster analysis (Figure 4 and Appendix A), suggesting that these specific TFs may regulate the expression of CDT genes. In addition, when low-stringency orthology analyses were carried out on the TF-encoding genes of the four species, we found that specific TF families were shared among two or three species (Table 5), suggesting a conserved role of these TFs among these cereal species. 

To validate these findings, we used the transcriptomic data related to an independent experiment of osmotic stress on two contrasting rice genotypes [28] and constructed a GCN (Figure 5A). Topological parameters derived from the analysis of the GCN local properties are commonly used for node ranking to identify essential genes in the network, as well as co-expressed functional modules [75]. Thus, to identify TFs that could have a crucial role in the regulation of CDT genes, the rice TF genes associated with photosynthesis and SGR clusters were compared to the TF genes displaying both high node degrees in the GCN and high expression correlations with the selected CDT genes (Figure 5B, Appendix A).

### 3.4. Rice Transcription Factors Involved in the Positive Regulation of CDT Photosynthesis Genes 

Fourteen TF genes, which were strongly positively and negatively correlated to *PsaO* and *SGR*, respectively, were identified as putative regulators of the photosynthesis-related genes. In particular, genes belonging to the nuclear factor Y (NF-Y or NF-YB/NF-YC) and COSTANS-like (CO-like) groups attracted our attention. NF-Y is a heterotrimeric TF that binds CCAAT-box elements in eukaryotic promoters and is conserved from yeast to mammals [76]. NF-Y genes have been suggested to play a role in acclimatization strategies for abiotic stress tolerance [77]. In rice, the down-regulation of *OsHAP3A* (NF-YB) results in reduced chlorophyll content in the leaves, degenerate chloroplasts, and a marked reduction in the mRNA level of the light-harvesting chlorophyll a/b-binding protein gene [78]. CO-like proteins were shown to interact directly with several NF-YB/NF-YCs to impart DNA sequence specificity to the histone fold NF-YB/NF-YC dimer and efficiently regulate NF-CO target genes [79]. NF-Y and CO-like genes were present in the GCN group associated with photosynthetic genes, and some members of these classes had characteristics of hubs in the rice GCN, that is, the NF-YB Os03g0413000, the NF-YC Os03g0251350, and CO-like Os03g0711100. Cis-elements that can be directly bound by NF-Y and NF-CO complexes are present in several CDT photosynthetic genes, including *PsaO, PsaK*, and *LHCB* (Appendix A). These NF-Y and CO-like TFs may therefore represent important TFs in the regulatory network underlying the promotion of photosynthesis vs. senescence in the establishment of drought stress tolerance. 

Among the relevant TFs in the photosynthetic cluster and GCN, we found the bHLH TF OsPIL-13 (Os03g0782500) and the bZIP TF OsbZIP13 (Os02g0128200). OsPIL13 (also called OsPIL1) plays an important role in both leaf senescence [80] and drought tolerance [81]. Recently, OsPIL13 was found to promote chlorophyll biosynthesis [82]. Under drought, OsPIL13 acts as a key regulator of reduced internode elongation in rice under drought, and thus, may be important for morphological stress adaptation under drought conditions [81]. *OsbZIP13* has been recognized as a major transcriptional regulator in the metabolic adjustments of rice under drought stress [83].

### 3.5. Rice Transcription Factors Involved in the Positive Regulation of Leaf Senescence 

Twenty TF genes were strongly positively and negatively correlated to *SGR* and *PsaO*, respectively. These TFs can be positive regulators of leaf senescence. Among them, *OsNAP*, *OsbZIP60* (Os07g0644100), and *MCB2MYB* (Os01g0863300) have been previously identified as candidate network hubs involved in the metabolic adjustments of rice under drought stress, as reported for *OsbZIP13* [83]. This confirms the strong interconnection of the two identified pathways in drought response. *OsNAP* (Os03g0327800) is well known for its pivotal role in the induction of leaf senescence and in connecting abscisic acid and leaf senescence [84,85]. *OsNAP* has been also reported to be associated with JA biosynthesis [86].

Among the 20 putative regulators of leaf senescence, *OsERF48* (Os08g0408500, also named *OsDRAP1* or *OsLG3*), has been well characterized for its ability to confer drought tolerance in rice plants when overexpressed [87,88]. In addition, it is located in a QTL for drought tolerance [89], and a natural allele of this gene, which was isolated from a tolerant rice accession, is able to confer drought tolerance by inducing an enhanced ROS scavenging response [90]. 

The cis-element analysis (Appendix A) showed that the *SGR* upstream sequence was enriched with putative binding sites for MYB, NAC, and ABRE TFs. This finding strengthened the hypothesis of direct regulation of the expression of *SGR* and other genes by the described TFs. The positive regulation of OsNAP on *SGR* has been well documented [84,85]. However, the role of the other identified TFs is not known, and correlation data may not discriminate between the negative and positive activity of TFs. Indeed, ERF, bZIP, and MYB genes are able to both positively and negatively regulate downstream genes [21,91,92,93]. Hence, OsbZIP13 can act as a positive regulator of the photosynthesis-related genes or a negative regulator of *SGR*. Similarly, OsbZIP60, MCB2MYB, and OsERF48 may induce *SGR* expression or repress the expression of photosynthesis-related genes. Further analyses are needed to reveal the exact role of these TFs. 

### 3.6. Putative Orthologs of Rice Transcription Factors Are Involved in the Regulation of CDT Genes in Brachypodium, Barley, and Maize

We detected 18 candidate TF orthology groups potentially involved in the regulation of drought tolerance genes (Table 5). Interestingly, some rice TF genes, which we have described above as master regulators of gene expression regulation, were present in these orthology groups. 

A bHLH group, which was related to photosynthesis, consisted of *OsPIL13* and its putative orthologs, Bradi1g06670 and Zm00001d034298. Another photosynthesis-related group found in Brachypodium, rice, and maize consisted of *OsbZIP13* and its putative orthologs Bradi3g02730, Zm00001d053967, and Zm00001d015118. The orthology analysis has also shown that the Bradi1g60030 is the putative ortholog of the NF-YB Os03g0413000, mentioned above as a possible master regulator of photosynthesis-related genes. 

Regarding leaf senescence regulation, an orthology group consisted of *OsERF48* and its putative orthologs Bradi3g35560 and Zm00001d032295. Moreover, an HD-ZIP orthology group (Bradi5g17170, HORVU2Hr1G092710, Zm00001d002799, and Zm00001d025964) was identified in Brachypodium, barley, and maize. The maize gene Zm00001d025964 was previously described as a candidate gene associated with leaf senescence [94]. The putative rice ortholog of these *HD-ZIP* genes is *OsHOX22* (Os04g0541700), which is able to confer tolerance to drought and salt through an ABA-mediate mechanism when overexpressed [95]. *OsHOX22* has been found to be differently regulated in sensitive and tolerant rice genotypes under both drought [96] and chilling [97] stresses. Conversely, a further study reported a similar up-regulation in both sensitive and tolerant cultivars under osmotic stress [98]. This data is consistent with the present analysis, where *OsHOX22* resulted in up-regulated DEG with similar expression profiles between the two genotypes. This behaviour could be due to the presence of different alleles in the analysed rice genotypes. In sum, *OsHOX22* was strongly negatively correlated to *PsaO* in the GCN (Appendix A), confirming a role in the regulation of drought-related expression.

Finally, a bHLH orthology group, which is related to the *SGR* clusters, consisted of the three genes Bradi2g00730, HORVU3Hr1G000170, and Os01g0108400, which have not been characterized yet, and thus, may represent novel candidates for drought tolerance in these cereal crops.

In conclusion, our analysis revealed the importance of the balance between leaf senescence and the maintenance of photosynthesis in drought tolerance and suggested that the fine-tuning of this balance through gene expression modulation by specific TFs can be the key to improving the ability of cereal crops to tolerate drought stress. The proposed model is presented in Figure 6.

The accuracy in the dataset selection, the comparison between the tolerant and sensitive genotypes, the combination of gene expression information across species, and the in-house-developed bioinformatics pipeline allowed us to identify potential drought tolerance core genes with high evolutionary conservation in cereals.

## 4. Materials and Methods

### 4.1. Selection of Published RNA-Seq Studies

Scientific papers about the transcriptomic analysis of tolerant response to drought in Gramineae species were searched in the Scopus and Google Scholar databases. The studies selected for this meta-analysis were selected based on the following criteria: (i) comparing the responses of two contrasting genotypes (one tolerant, one sensitive) to drought stress; (ii) drought stress performed in soil by withholding water for medium to long periods (several days, not hours) during the plant vegetative stage; (iii) RNA isolated from pooled samples of leaves; (iv) raw RNA-Seq data available in public repositories. These criteria resulted in the selection of 4 datasets belonging to 4 different species—*B. distachyon* [30], *H. vulgare* [29], *O. sativa* [32], and *Z. mays* [31]. If different time point series were carried out in the same study, a single time point was selected so that the drought severity was comparable with the other considered datasets. Table 1 summarizes the main features of the materials used for this study.

A defined code was used to standardize the sample identifiers. The first part of the name indicated the species (“Bd” = *B. distachyon*; “Hv” = *H. vulgare*; “Os” = *O. sativa*; “Zm” = *Z. mays*), and the part following it was referred to as the drought stress-related phenotype of the cultivar (“sen” = sensitive; “tol” = tolerant). The last letter indicates the growth condition (C = control; S = stressed), and the final number represents the biological replicate (1, 2, or 3).

### 4.2. RNA-Seq Data Handling and Gene Expression Quantification

The pipeline for RNA-Seq data handling was similar to the one that we used in a previous study [35]. Briefly, FastQC 0.11.9 [99] was used to assess the RNA read qualities of the libraries summarized in Table 1, while Trimmomatic 0.39 [100] was used to filter out the adaptor sequences and the low-quality bases. The filtered RNA reads were then mapped to the respective reference genome using HiSat2 2.2.1 aligner [101] with default parameters. The most recent reference genome assemblies for the species used in this study were retrieved from the EnsemblPlants website (http://plants.ensembl.org/, accessed on 30 September 2021) on 7 February 2020. RNA reads of Brachypodium, barley, maize, and rice were mapped to the *B. distachyon* v3.0 [102], *H. vulgare* IBSC v2 [103], *Z. mays* B73 RefGen v4 [104], and *O. sativa* ssp. *japonica* IRGSP-1.0 [105] genome assemblies, respectively.

Finally, read counts were generated from alignment files with featureCounts software, part of the Subread package 1.6.2 [106], with default parameters, basing on ‘exon’ feature into ‘gene_id’ meta-feature of gtf annotation files retrieved from the EnsemblPlants website (http://plants.ensembl.org/, accessed on 30 September 2021) on 7 February 2020. The annotation files used for the reads counts were *B. distachyon* v3.0.46, *H. vulgare* IBSC v2.46, *Z. mays* B73 RefGen v4.46, and *O. sativa* IRGSP 1.0.46. Multi-mapping and multi-overlapping reads were not counted.

### 4.3. Differential Expression Analysis and Orthology Study

DE analyses comparing the stressed and control samples were carried out separately for each dataset using Bioconductor EdgeR 3.28.1 [107] in the R environment. EdgeR was used to (i) filter out the genes that were not expressed or poorly expressed (a gene was considered “active” if the reads per million mapped to that gene were >1 in at least two libraries), (ii) normalize the RNA libraries, and (iii) perform the differential expression analysis with the likelihood ratio test comparing the treated (stressed) samples to the control (not stressed) ones. The log_2_ fold change (LFC) of expression between the treated and control samples was calculated with EdgeR, whose computing approach fits a negative binomial generalized linear model (GLM) to the read counts for each gene. The genes with a resulting false discovery rate (FDR) smaller than 0.05 were considered DEGs. No LFC cut-off was used for DEG identification.

In order to identify the orthologous rice genes for the other three species, a complete list of orthologous genes corresponding to *O. sativa* from Ensembl plants was collected. An in-house Perl script was run to parse the corresponding ortholog gene from *B. distachyon*, *Z. mais*, and *H. vulgare*.

### 4.4. Transcription Factors Annotation and Orthology

Manually-curated TF annotations for the four cereal species were carried out on the DEG genes for each species, combining TF annotations already present in the plant transcription factor database (Plant TFDB 3.0; [108]), the gene description from the reference genome assemblies (*B. distachyon* v3.0.46, *H. vulgare* IBSC v2.46, *Z. mays* B73 RefGen v4.46, and *O. sativa* IRGSP 1.0.46), and the best *A. thaliana* orthology hits. TF classes were annotated according to the plant TFDB classification. For the orthology analysis, the gene IDs of the *B. distachyon*, *H. vulgare*, and *Z. mays* TFs in the negatively correlated clusters related to photosynthesis and senescence were used for searching the corresponding orthologues (to a maximum of 6) in rice using the Ensembl Biomart tool in the plant platform [109]. The resulting rice IDs were used in a VENN analysis to identify those that were common to more species.

### 4.5. Protein–Protein Interaction Network Analysis

Protein–protein interaction (PPI) networks were analysed with STRING v11 [110], using a combined score threshold of 0.7.

### 4.6. K-Means Cluster Analysis 

The pipeline for the k-means cluster analysis was performed according to Testone et al. (2019) with some modifications. Briefly, normalized counts from EdgeR were converted to transcripts per kilobase million (TPM) for each species with the following procedure: the read counts were divided by the length of each gene in kilobases to obtain reads per kilobase (RPK); all the RPK values in a sample were counted and this number was divided by 1,000,000 to obtain the “per million” scaling factor; the RPK values were divided by the “per million” scaling factor to obtain the TPM values. TPM means values of TF DEGs and of the 69 CDT genes were used for cluster analysis. The TPM means from each sample were log-transformed using log_2_(x + 1) for data normalization. Each gene was indexed according to the species (Bd: *B. distachyon;* Hv: *H. vulgare;* Os: *O. sativa;* Zm: *Z. mays*). The optimum number of clusters was determined as described by Testone et al. (2019) [58]. Data scaling, k-means clustering, and visualization were performed in R according to the methods of Morgan et al. [111].

### 4.7. Gene Co-Expression Network (GCN) Construction and Analysis

For the GCN construction, we used the RPKM gene expression values of CDTs and differentially expressed TF genes from a previous experiment of PEG-mediated osmotic stress in rice [28]. The data were log-transformed using log_2_(x + 1) for normalization, and Pearson pairwise correlation analysis was conducted across the selected samples using the “corrplot” and “hclust” R packages of R software [112]. Significant correlations (*p*-value ≤ 0.05), with a Pearson’s correlation coefficient (r) ≥ |0.9| were used for the construction of co-expression networks and network analysis in the Cytoscape software platform v. 3.5.1 [59].

### 4.8. Analysis of the Cis-Regulatory Elements in the CDT Promoters

Cis-regulatory motif analysis was carried out using the regulatory sequence analysis tools (RSAT) web platform [113]. The 3000 bp regions upstream of the translation start sites of the 69 CDT rice genes were retrieved from the *O. sativa* Japonica IRGSP-1 genome using the RSAT retrieve sequence tool. The sequences were then analysed for the presence of 16 consensus motifs for NF-Y, NF-CO, NAC, bZIP, MYB, and ERF TFs [79,114,115] (Appendix A). The total/average values in the 3000 bp of each motif were calculated by dividing the total number of motifs found in the 3000 bp promoters of all rice annotated genes by the number of rice annotated genes. This value was then used to identify CDT genes enriched in a particular motif by dividing the number of motif hits by the average motif occurrence in the 3000 bp regulatory regions of rice genes. 

## Figures and Tables

**Figure 1 ijms-22-13062-f001:**
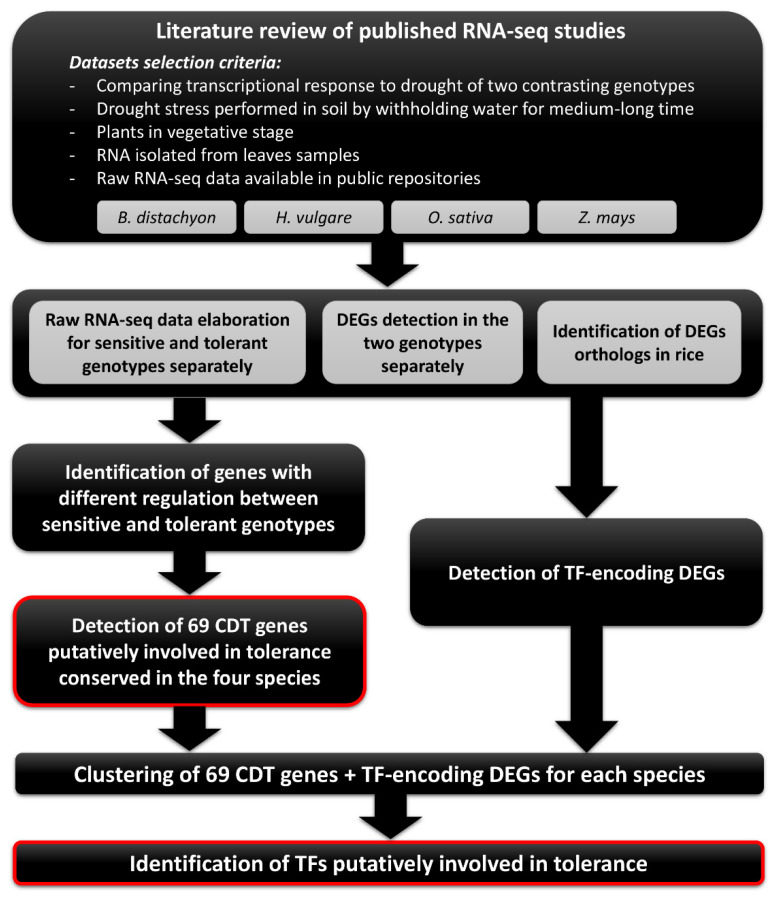
Overall workflow used in this study for the identification of genes putatively involved in drought tolerance in cereal crops, as well as upstream transcription factors involved in their regulation. CDT genes: Conserved Drought-tolerance Related genes.

**Figure 2 ijms-22-13062-f002:**
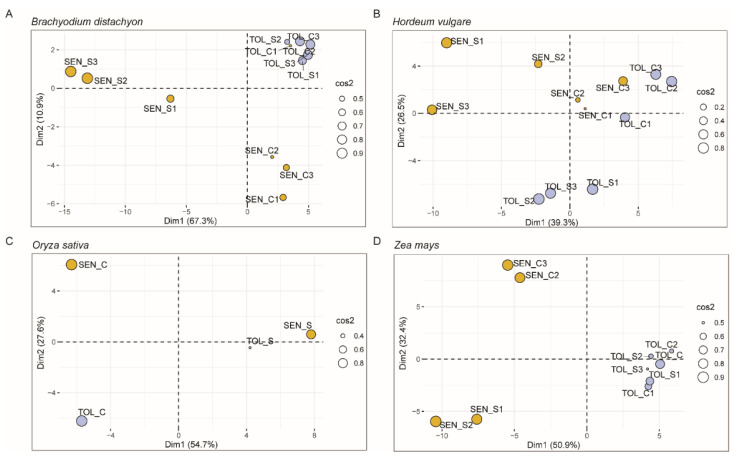
Principal component analysis (PCA) biplots of the four RNA-Seq datasets using the TPM expression values of the CDT genes of *B. distachyon* (**A**), *H. vulgare* (**B**), *O. sativa* (**C**), and *Z. mays* (**D**). Samples from the sensitive genotypes are in yellow, whereas those from the tolerant genotypes are in light blue. Squared cosine (cos2) indicates the contribution of each sample to a given observation; the bigger the circle is, the higher the contribution is. TOL_C: biological replicates of the tolerant genotypes that were used as non-stressed control in the experiments; TOL_S: biological replicates of the tolerant genotypes under drought stress; SEN_C: biological replicates of the sensitive genotypes that were used as non-stressed controls; SEN_S: biological replicates of the sensitive genotypes under drought stress.

**Figure 3 ijms-22-13062-f003:**
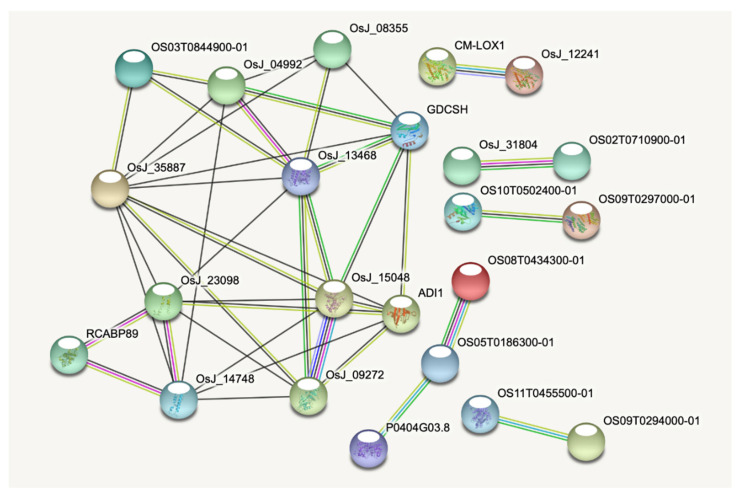
Protein–protein interaction (PPI) network using the 69 rice proteins. Only proteins involved in interactions with a high confidence (combined score higher than 0.7) were shown. For each shown protein, the related rice gene ID is indicated in parentheses: OS03T0844900-01 (Os03g0844900); OsJ_04992 (Os02g0101500); OsJ_08355 (Os02g0744000); OsJ_35887 (Os12g0420200); OsJ_13468 (Os04g0234600); GDCSH (Os10g0516100); OsJ_23098 (Os07g0148900); OsJ_15048 (Os04g0459500); ADI1 (Os08g0104600); RCABP89 (Os03g0592500); OsJ_14748 (Os04g0414700); OsJ_09272 (Os03g0129300); CM-LOX1 (Os08g0508800); OsJ_12241 (Os03g0700700); OsJ_31804 (Os10g0462900); OS02T0710900-01 (Os02g0710900); OS10T0502400-01 (Os10g0502400); OS09T0297000-01 (Os09g0297000); OS08T0434300-01 (Os08g0434300); OS05T0186300-01 (Os05g0186300); P0404G03.8 (Os06g0486800); OS11T0455500-01 (Os11g0455500); OS09T0294000-01 (Os09g0294000).

**Figure 4 ijms-22-13062-f004:**
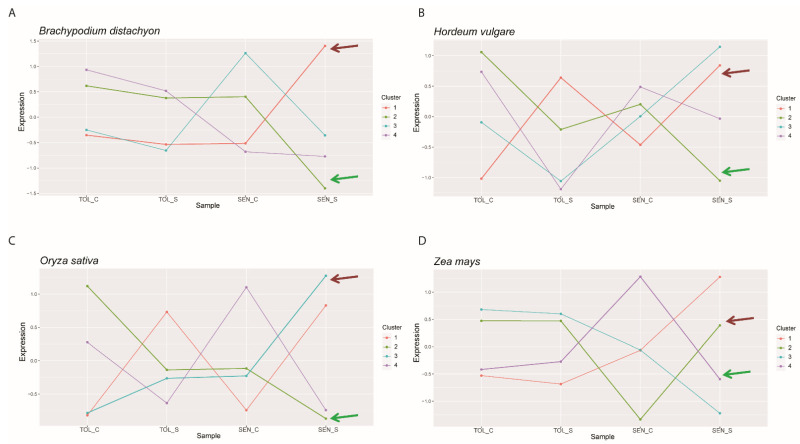
K-means cluster analysis of gene expression of TF-encoding DEGs and CDT genes in the four cereal datasets. Distance matrix for k-means clustering was calculated by Euclidean similarity measurement and using centred Pearson’s correlation as the distance metric, resulting in four gene clusters for each species: *B. distachyon* (**A**), *H. vulgare* (**B**), *O. sativa* (**C**), and *Z. mays* (**D**). Expression refers to log2+1 normalized TPM values that were then scaled to identify clusters of genes that share similar expression profiles rather than similar expression levels. The lines in the cluster plots represent the centroids in each cluster, around which genes with similar expression profiles are associated. The negatively correlated clusters harbouring *SGR* genes or photosynthesis-related genes (the genes *CA1/PsaO/PsaK/LHCB* for Brachypodium, barley, and rice, the genes *CA1/PsaO//LHCB* for maize) are indicated by brown or green arrows, respectively. TOL_C: gene expression of tolerant genotypes that were used as non-stressed controls in the experiments; TOL_S: gene expression of tolerant genotypes under drought stress; SEN_C: gene expression of sensitive genotypes that were used as non-stressed controls; SEN_S: gene expression of sensitive genotypes under drought stress.

**Figure 5 ijms-22-13062-f005:**
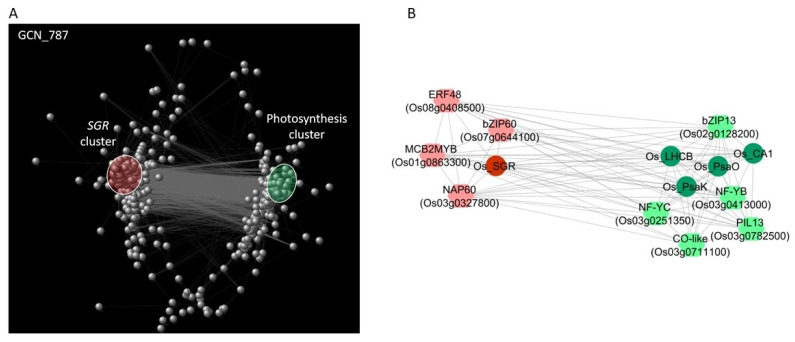
(**A**) Gene co-expression network (GCN) created with Cytoscape related to an independent osmotic stress experiment [28]. The nodes indicate the 787 genes belonging to the network, that is, CDT genes and DEG–TFs. Grey lines represent significant (*p*-value ≤ 0.05) gene expression correlations higher than |0.9|. The two negatively correlated groups of the GCN, which correspond to the two negatively correlated clusters shown in Figure 4C and are related to photosynthesis genes and SGR, are highlighted in green and red, respectively. (**B**) Graphical representation of the principal GCN hub genes related to photosynthesis and SGR pathways that are discussed in the text. Grey lines represent significant (*p*-value ≤ 0.05) gene expression correlations higher than |0.9|.

**Figure 6 ijms-22-13062-f006:**
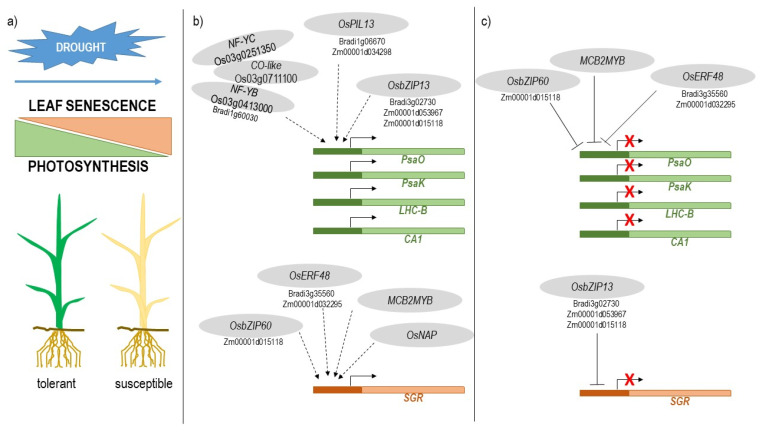
Proposed model for the differentiation of tolerant and sensitive response to drought stress in cereals. (**a**) The balance between induction of leaf senescence and maintenance of photosynthesis during drought is crucial for tolerance and is obtained through fine-tuning of gene expression by specific transcription factors (TFs). (**b**,**c**) The putative rice TFs involved in the regulation of genes related to photosynthesis (*Psao*, *PsaK*, *LHCB,* and *CA1*) and leaf senescence (*SGR*) are represented. Some of them may act as positive (**b**) or negative (**c**) regulators. When identified, putative barley and maize orthologues of rice TF-encoding genes are indicated.

**Table 1 ijms-22-13062-t001:** Main features of RNA datasets used for meta-analysis. References, cultivars, biological replicates, drought stress modalities, sequencing techniques, and accessions from online repository BioProject were reported.

	*B. distachyon*	*H. vulgare*	*O. sativa*	*Z. mays*
Reference	[30]	[29]	[32]	[31]
Sensitive accession	KOZ1	Scarlett	HanFengB	MO17
Tolerant accession	ABR8	SBCC073	HuHuan2B	YE8112
Biological replicates	3	3	Pooled	2–3
Drought time	8 days	30 days	20 days	7 days
Sequencer	HiSeq 2000	HiSeq 2000	HiSeq 2500	Hiseq Xten
Reads length	100 nt paired-end	101 nt paired-end	126 nt paired-end	150 nt single-end
BioProject accession	PRJNA524106	PRJEB12540	PRJNA306542	PRJNA397964

**Table 2 ijms-22-13062-t002:** Numbers of predicted genes in the reference assembly for each species, of active genes and of down- and up-regulated DEGs for each experiment.

	*B. distachyon*		*H. vulgare*		*O. sativa*		*Z. mays*	
	Sensitive	Tolerant	Sensitive	Tolerant	Sensitive	Tolerant	Sensitive	Tolerant
Predicted genes	35,118	35,118	43,050	43,050	39,978	39,978	45,953	45,953
Active genes	19,404	18,411	16,092	15,773	18,449	18,285	20,094	21,048
Down-reg DEGs	3551	61	287	1261	1634	1360	2915	583
Up-reg DEGs	3145	234	448	653	1650	1134	2129	348
Total DEGs	6696	295	735	1914	3284	2494	5044	931

**Table 3 ijms-22-13062-t003:** Genes putatively involved in the tolerant response to drought stress. The numbers of genes belonging to the eight classes were reported for each dataset.

Class	*B. distachyon*	*H. vulgare*	*O. sativa*	*Z. mays*
sen+	3125	313	942	2026
sen−	3340	168	901	2722
tol+	38	543	421	251
tol−	26	1117	632	384
∆LFC > 1	16	12	88	56
∆LFC < −1	4	13	40	9
sen−/tol+	185	1	19	77
sen+/tol−	9	26	14	83
TOTAL	6743	2193	3057	5608

**Table 4 ijms-22-13062-t004:** Rice Conserved drought tolerance-related (CDT) genes. Where defined, the gene names and the pathways in which these genes are involved are shown.

n.	RAP ID	Gene Name	Related Pathway
Os_1	Os01g0111100	uncharacterized gene	
Os_2	Os01g0117300	*OsRLCK17*	signalling
Os_3	Os01g0227800	uncharacterized gene	
Os_4	Os01g0551000	uncharacterized gene	
Os_5	Os01g0639900	*OsCA1*	PHOTOSYNTHESIS (CO_2_ fixation)
Os_6	Os01g0733500	*RD22*	ABIOTIC STRESS RESPONSE
Os_7	Os01g0803200	*OsCYS1*	cysteine proteinase inhibitors
Os_8	Os02g0101500	*Peroxisomal hydroxypyruvate reductase*	
Os_9	Os02g0465900	*ChaC-like*	TRANSPORT
Os_10	Os02g0626400	*PAL2-2*	SECONDARY METABOLISM
Os_11	Os02g0710900	*HSP70—OsBiP*	ABIOTIC STRESS RESPONSE
Os_12	Os02g0744000	uncharacterized gene	
Os_13	Os02g0783625	*lysine ketoglutarate reductase*	
Os_14	Os02g0815700	uncharacterized gene	
Os_15	Os03g0114800	*OsCd1*	TRANSPORT
Os_16	Os03g0129300	*G3PDH*	PHOTOSYNTHESIS (Calvin Cycle)
Os_17	Os03g0192700	*Inositol-3-phosphate synthase 1*	
Os_18	Os03g0238300	*inositol-1,4,5-trisphosphate 5-phosphatase*	
Os_19	Os03g0336000	uncharacterized gene	
Os_20	Os03g0592500	*LHCB*	PHOTOSYNTHESIS (photosystem assembly)
Os_21	Os03g0700700	*OsLOX8*	JA SIGNALLING
Os_22	Os03g0744700	uncharacterized gene	
Os_23	Os03g0844900	uncharacterized gene	
Os_24	Os03g0849800	*Glycosyltransferase*	
Os_25	Os03g0850400	*aspartate kinase*	
Os_26	Os04g0234600	*SBPase*	PHOTOSYNTHESIS (Calvin Cycle)
Os_27	Os04g0390100	*OsHIPP29*	TRANSPORT
Os_28	Os04g0414700	*PsaO*	PHOTOSYNTHESIS (photosystem assembly)
Os_29	Os04g0459500	*G3PDH*	PHOTOSYNTHESIS (Calvin Cycle)
Os_30	Os04g0496000	uncharacterized gene	
Os_31	Os04g0508800	uncharacterized gene	
Os_32	Os04g0618200	*UDP-arabinose 4-epimerase 2*	
Os_33	Os05g0186300	*OsNADP-ME3*	PHOTOSYNTHESIS (Calvin Cycle)
Os_34	Os05g0429500	*Dienelactone hydrolase*	
Os_35	Os05g0439400	*OsPUB44*	*E3 ubiquitin ligase*
Os_36	Os05g0453300	uncharacterized gene	
Os_37	Os05g0468900	uncharacterized gene	
Os_38	Os05g0501700	uncharacterized gene	
Os_39	Os05g0535900	*IQ calmodulin-binding motif family protein*	
Os_40	Os06g0129400	*OsSPX-MFS3*	CELLULAR TRANSPORT
Os_41	Os06g0486800	*FDH* (Formate dehydrogenase)	CELLULAR TRANSPORT
Os_42	Os06g0528600	*Aminopropyl transferase*	
Os_43	Os06g0716000	uncharacterized gene	
Os_44	Os07g0148900	*Psak*	PHOTOSYNTHESIS (photosystem assembly)
Os_45	Os07g0524900	uncharacterized gene	
Os_46	Os08g0104600	*OsFd1*	OXIDATIVE STRESS RESPONSE
Os_47	Os08g0141300	*OsVST1*	root system architecture
Os_48	Os08g0141400	*NADH-ubiquinone oxidoreductase*	
Os_49	Os08g0364900	uncharacterized gene	
Os_50	Os08g0434300	*malate dehydrogenase*	
Os_51	Os08g0485900	uncharacterized gene	
Os_52	Os08g0508800	*OsHI-LOX*	JA SIGNALLING
Os_53	Os09g0294000	*Aspartate kinase-homoserine dehydrogenase*	
Os_54	Os09g0297000	*Ferrochelatase-1*	
Os_55	Os09g0426800	*OsGL1-1*	leaf wax synthesis
Os_56	Os09g0438000	*OsNOX7*	OXIDATIVE STRESS RESPONSE
Os_57	Os09g0438100	uncharacterized gene	
Os_58	Os09g0532000	*SGR*	LEAF SENESCENCE
Os_59	Os10g0462900	*OsHsp60-1*	ABIOTIC STRESS RESPONSE
Os_60	Os10g0502400	*OsHEMA*	LEAF SENESCENCE
Os_61	Os10g0516100	*OsGDCH*	LEAF SENESCENCE
Os_62	Os10g0530500	*glutathione S-transferase*	OXIDATIVE STRESS RESPONSE
Os_63	Os11g0145200	*Indole-3-acetate beta-glucosyltransferase*	AUXIN SIGNALLING
Os_64	Os11g0455500	*OsSAHH*	SECONDARY METABOLISM
Os_65	Os11g0520500	uncharacterized gene	
Os_66	Os12g0115700	*CHI*	SECONDARY METABOLISM
Os_67	Os12g0420200	*CSP41b*	OXIDATIVE STRESS RESPONSE
Os_68	Os12g0428000	*glycosyl hydrolase*	
Os_69	Os12g0534100	uncharacterized gene	

**Table 5 ijms-22-13062-t005:** Candidate TF genes related to drought tolerance across the four species.

*B. distachyon* Gene ID	*H. vulgare* Gene ID	*O. sativa* Gene ID	*Z. mays* Gene ID	Corresponding Clusters	TF Class
Bradi1g06670	-	Os03g0782500	Zm00001d034298	photosynthesis	bHLH
Bradi2g00730	HORVU3Hr1G000170	Os01g0108400	-	*SGR*	bHLH
Bradi2g60970	-	Os01g0952800	-	photosynthesis	bHLH
Bradi3g02730	-	Os02g0128200	Zm00001d053967 Zm00001d015118	photosynthesis	bZIP
Bradi3g57960	-	-	Zm00001d018178	*SGR*	bZIP
	-	Os07g0644100	Zm00001d007042	*SGR*	bZIP
Bradi4g02570	-	Os12g0601800	-	photosynthesis	bZIP
Bradi1g11310	-	-	Zm00001d033719 Zm00001d013443	photosynthesis	CO-like
Bradi3g35560	-	Os08g0408500	Zm00001d032295	*SGR*	ERF
Bradi4g29380	-	Os09g0379600		*SGR*	HD-ZIP
Bradi5g17170	HORVU2Hr1G092710	-	Zm00001d002799Zm00001d025964	*SGR*	HD-ZIP
Bradi3g16400	-	Os08g0159500	-	photosynthesis	LBD
Bradi2g16120	-	Os05g0567600	-	photosynthesis	MYB
Bradi3g16515	HORVU7Hr1G070870	-	-	photosynthesis	MYB
Bradi3g52807	-	-	Zm00001d017782	photosynthesis	MYB
-	HORVU5Hr1G099390	-	Zm00001d013151	*SGR*	NAC
Bradi1g60030	-	Os03g0413000	-	photosynthesis	NF-YB
-	-	Os06g0145800	Zm00001d036148	photosynthesis	Whirly

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
