# Peer review of "A Comparative Transcriptomic Meta-Analysis Revealed Conserved Key Genes and Regulatory Networks Involved in Drought Tolerance in Cereal Crops"

_ijms, 2021, doi:10.3390/ijms222313062_

Round 1
Reviewer 1 Report
In this research article, the authors aim to identify genes that may represent key players for conferring drought tolerance in different Gramineae species, and four different species including three C3 cereal plants and one C4 cereal plants were selected. There are 69 genes which potentially involved in drought tolerance for all the four species were identified. Interestingly, using K-means clustering analysis of the drought response between tolerant and sensitive genotypes, the author indicated a prominent role of leaf senescence-related mechanisms, which may provide new strategy for future drought tolerant crop breeding. Here, a few comments need to be considered in this paper.
- There are no row number in the manuscript for convenient peer-review.
- The title of the manuscript is too long, up to 30 words, please reorganize the sentence to a short and refined one.
- A NAC transcription factor RD26 which proven play central role in regulating drought stress response (Gupta et al., Science 2020, 368:266–269) is also identified as a key regulator in leaf senescence (Wang et al., The Plant Cell 2021, 33: 1594–1614). I wonder whether this TF is one of the 69 genes.
- In the Discussion section 3.3, ‘Senescence-Related Genes Play a Major Role in Drought Tolerance’. As a highlight part of this manuscript, the authors need to strengthen this paragraph. It’s better to change the internal logical organization to ‘Drought stress response gene regulates leaf senescence signaling pathway’. Because SGR displayed in Figure 6 is more likely a chlorophyll catabolic gene (CCG), acts as downstream gene and targeted by many leaf senescence key regulators.
- In page 11, title of table 5, ‘The genes shared among three species are in bold’. However, there are no words in bold in this table.
- It’s better to add percentage (%) in Figure 5 for readability.
- Please standardize the Latin name writing, ‘Brachypodium distachyon’ for first time mentioned, then using ‘ distachyon’ for all the others (Page 3, Page 5, Page10, Page 11, etc.).
- In page 15, Section 3.5, line 16 to 18, ‘These observations confirmed that our analysis could effectively … of leaf senescence mechanisms for a successful response to drought’. Please remove this sentence to ‘Discussion’ part.
- In page 20, Reference 23, all first letter of the words are capitalized. Similar situation of Reference 32, 33, 46, 58, 66.
Author Response
Reviewer 1
In this research article, the authors aim to identify genes that may represent key players for conferring drought tolerance in different Gramineae species, and four different species including three C3 cereal plants and one C4 cereal plants were selected. There are 69 genes which potentially involved in drought tolerance for all the four species were identified. Interestingly, using K-means clustering analysis of the drought response between tolerant and sensitive genotypes, the author indicated a prominent role of leaf senescence-related mechanisms, which may provide new strategy for future drought tolerant crop breeding. Here, a few comments need to be considered in this paper.
We thank the reviewer for having appreciated our study and for the pertinent observations that are helping us to improve the manuscript. We are going to explain, point by point, the details of the revisions to the manuscript.
- There are no row number in the manuscript for convenient peer-review.
The row numbers were not visible in the manuscript because the IJMS template for manuscript submission doesn’t include them. Anyway, in the revised version, line numbering was added as asked by the Reviewer.
- The title of the manuscript is too long, up to 30 words, please reorganize the sentence to a short and refined one.
The manuscript title was shortened and refined. The new title is “A Comparative Transcriptomic Meta-Analysis Revealed Conserved Key Genes and Regulatory Networks Involved in Drought Tolerance in Cereal Crops”
- A NAC transcription factor RD26 which proven play central role in regulating drought stress response (Gupta et al., Science 2020, 368:266–269) is also identified as a key regulator in leaf senescence (Wang et al., The Plant Cell 2021, 33: 1594–1614). I wonder whether this TF is one of the 69 genes.
We searched for the putative ortholog of AtRD26 (AT4G27410) in Ensemble plant, but the analysis underlined that 61 species don't have any orthologue with AT4G27410 (http://plants.ensembl.org/Arabidopsis_thaliana/Gene/Compara_Ortholog?g=AT4G27410;r=4:13707192-13709170#list_no_ortho). Among them, cereal species are present, including rice, Brachypodium, maize and barley. It is known that many TF genes differently specialized between monocots and dicots, and even among species within these two clades.
- In the Discussion section 3.3, ‘Senescence-Related Genes Play a Major Role in Drought Tolerance’. As a highlight part of this manuscript, the authors need to strengthen this paragraph. It’s better to change the internal logical organization to ‘Drought stress response gene regulates leaf senescence signaling pathway’. Because SGR displayed in Figure 6 is more likely a chlorophyll catabolic gene (CCG), acts as downstream gene and targeted by many leaf senescence key regulators.
The title of this paragraph (3.4 in the new version) has been changed in “The Balance between Induction of Leaf Senescence and Maintenance of Photosynthesis Plays a Major Role in Drought Tolerance”. In the new version, to strengthen this paragraph we added some sentences, and related references, about the importance of suppression of drought-induced leaf senescence to improve drought-tolerance.
- In page 11, title of table 5, ‘The genes shared among three species are in bold’. However, there are no words in bold in this table.
Maybe the formatting of the Table 5 got lost during the manuscript submitted version preparation. Thanks for pointed it out. Now the shared genes among the three species are in written in bold.
- It’s better to add percentage (%) in Figure 5 for readability.
Since we added a further figure 5 related to new analyses, we added percentages and moved the previous Figure 5 to Supplementary Material.
- Please standardize the Latin name writing, ‘Brachypodium distachyon’ for first time mentioned, then using ‘distachyon’ for all the others (Page 3, Page 5, Page10, Page 11, etc.).
Latin names are now standardized: for all the species, the first mention was written extensively, while for the following ones we used the contracted form (e.g. B. distachyon).
- In page 15, Section 3.5, line 16 to 18, ‘These observations confirmed that our analysis could effectively … of leaf senescence mechanisms for a successful response to drought’. Please remove this sentence to ‘Discussion’ part.
This sentence has been removed.
- In page 20, Reference 23, all first letter of the words are capitalized. Similar situation of Reference 32, 33, 46, 58, 66.
Mentioned references (as well as other references with the first letter of the words capitalized (i.e., 4, 17, 55, 62, 79, 89, 117) were fixed.

Reviewer 2 Report
In this manuscript, the authors described a comparative transcriptomic meta-analysis related to four Gramineae species, and identified 69 genes that are potentially involved in drought tolerance for all the analysed species. I feel that the transcriptomic meta-analysis is too simple, too descriptive and too preliminary at this stage. The study is not very original and find the metabolism pathways of stress response, photosynthesis, chlorophyll biogenesis, secondary metabolism, jasmonic acid signalling, cellular transport that are largely conserved in Arabidopsis and other plants. Not sufficient mechanistic insight can be provided from the simple transcriptomic analysis in this study. Hence, the authors need to provide more functional validation of some key genes or transcription factors, which are not yet characterized and might be involved in drought tolerance.
Major comments:
- In Abstract: Some genes are not yet characterized and can be novel candidates for drought tolerance. Which genes?
- In Abstract: “In addition, we identified specific TFs that could regulate the expression of leaf senescence-related genes.” Here, the authors need to provide more functional validation of some key genes or transcription factors.
- In Introduction: the authors mainly described morphological, physiological and molecular mechanisms, such as stomatal closure, CO2 absorption and the genes and pathways. However, key transcription factor genes and metabolic pathways are not clearly described.
- In the Results: “total RNA was extracted from leaves of stressed and not-stressed plants of sensitive and tolerant accessions at vegetative stage and subsequently sequenced with HiSeq technology, and the corresponding RNA-Seq raw reads were available in public repositories.” should be in “Materials and Methods”.
- In the Results: “Data elaboration was carried out separately for the four datasets, and the results were then integrated to identify the genes whose drought-induced expression variation differed between tolerant and sensitive genotypes.” should be in “Materials and Methods”.
- “Normalization factors were calculated according to the respective library size, and normalized reads counts were used as input data for differential expression analyses of stressed vs control samples for both tolerant and sensitive accessions. The results of the eight differential expression analyses (2 genotypes, 4 species) are reported in Supplementary File S1 and graphically represented in Figure S1. The number of up- and down-regulated differentially expressed genes (DEGs) for each analysis, along with the number of active genes, is summarised in Table 2.” This description lacks focus.
- The authors need to provide more expression analysis and functional validation of the key genes.
- In Results, the authors only described the analysis of the Protein-Protein Interaction of key genes, but addressing important biological questions is left. Therefore, protein interaction experiments key genes must be provided.
- In Discussion, the authors should discuss more on the differences of key drought tolerance-related genes between the previous studies and this study.
- The discussion section needs to be rewritten. In Discussion, I feel that the authors did not clarify the relationships between the genes and drought tolerance.
Author Response
Reviewer 2
In this manuscript, the authors described a comparative transcriptomic meta-analysis related to four Gramineae species, and identified 69 genes that are potentially involved in drought tolerance for all the analysed species. I feel that the transcriptomic meta-analysis is too simple, too descriptive and too preliminary at this stage. The study is not very original and find the metabolism pathways of stress response, photosynthesis, chlorophyll biogenesis, secondary metabolism, jasmonic acid signalling, cellular transport that are largely conserved in Arabidopsis and other plants. Not sufficient mechanistic insight can be provided from the simple transcriptomic analysis in this study. Hence, the authors need to provide more functional validation of some key genes or transcription factors, which are not yet characterized and might be involved in drought tolerance.
We thank the reviewer for the pertinent observations that are helping us to improve the manuscript. We have modified the manuscript according to the suggestions. We are going to explain, point by point, the details of the revisions.
- In Abstract: Some genes are not yet characterized and can be novel candidates for drought tolerance. Which genes?
The genes not yet characterized that can be novel candidates for drought tolerance are 20, so they could not be mentioned in the Abstract paragraph. These genes were reported in detail in Table 4. Anyway, in order to increase the readability, we added the number of not yet characterized genes in the Abstract as well.
- In Abstract: “In addition, we identified specific TFs that could regulate the expression of leaf senescence-related genes.” Here, the authors need to provide more functional validation of some key genes or transcription factors.
This manuscript is focused on bioinformatic analyses, as described in the title, and the biological validation of genes of interest is beyond the scope of this work. It is frequent that meta-analyses do not report biological studies of the identified candidate genes (see for example: Sircar and Parekh 2019 https://doi.org/10.1371/journal.pone.0216068; Shaar-Moshe et al. 2015 DOI 10.1186/s12870-015-0493-6; Muthuramalingam et al. 2017 doi: 10.3389/fpls.2017.00759; Cohen and Leach 2019 https://doi.org/10.1038/s41598-019-42731-8). Anyway, we thank the reviewer for highlighting this criticism and added additional analyses to strengthen our findings. We validated the results with an independent experiment related to osmotic stress (Baldoni et al. 2016, DOI 10.1186/s12284-016-0098-1), that we successfully used for a previous meta-analysis related to abiotic stress tolerance in rice (Buti et al. 2019 doi:10.3390/ijms20225662) and that already identified three candidate genes that are in common with this work (as we mentioned in the Manuscript, see paragraph 3.3 and 3.4 of the first version and lines 583-587 and lines 600-603 of the revised version). We constructed a gene co-expression network using the data of the independent experiment (see section 2.8. “Gene co-expression network analysis of the 69 CDT genes and TF DEGs on an independent water stress experiment in rice”.) We then validated both the 69 genes involved in stress tolerance and many transcription factors putatively involved in their regulation, by GCN regulatory network analysis. In addition, we further strengthened the possible interactions between specific TFs and the 69 genes by analysing the regulatory cis-elements in the promoters of the 69 rice genes (see the paragraph 2.9. “Analysis of TF binding sites on the promoter sequences of the rice CDT genes”).
- In Introduction: the authors mainly described morphological, physiological and molecular mechanisms, such as stomatal closure, CO2 absorption and the genes and pathways. However, key transcription factor genes and metabolic pathways are not clearly described.
A description about the metabolic pathways involved in drought response (lines 67-92 of the revised version) and the key transcription factor genes (lines 104-125 of the revised version) has been added in the Introduction (see the PDF manuscript version with the change tracking), together with the related new references.
- In the Results: “total RNA was extracted from leaves of stressed and not-stressed plants of sensitive and tolerant accessions at vegetative stage and subsequently sequenced with HiSeq technology, and the corresponding RNA-Seq raw reads were available in public repositories.” should be in “Materials and Methods”.
The methods mentioned in this sentence were already described in details in Materials and Methods paragraph 4.1 but, since in IJMS papers the Materials and Methods section comes after the Results section, we briefly described the experimental design of the four studies we used for the meta-analysis also here. As the Reviewer pointed out, maybe the aim of our sentence was not clear enough, so we shortened it and added an explicit reference to the 4.1 paragraph of the manuscript.
- In the Results: “Data elaboration was carried out separately for the four datasets, and the results were then integrated to identify the genes whose drought-induced expression variation differed between tolerant and sensitive genotypes.” should be in “Materials and Methods”.
Since the methods mentioned in this sentence were described in details in paragraphs 4.2 and 4.3, we simply removed the sentence.
- “Normalization factors were calculated according to the respective library size, and normalized reads counts were used as input data for differential expression analyses of stressed vs control samples for both tolerant and sensitive accessions. The results of the eight differential expression analyses (2 genotypes, 4 species) are reported in Supplementary File S1 and graphically represented in Figure S1. The number of up- and down-regulated differentially expressed genes (DEGs) for each analysis, along with the number of active genes, is summarised in Table 2.” This description lacks focus.
The sentences mentioned by the Reviewer actually lacked focus. We deleted the parts already described in Materials and Methods section and tried to improve the clarity of the text.
- The authors need to provide more expression analysis and functional validation of the key genes.
Please refer to the answer at point 2 for the new analyses that have been provided.
- In Results, the authors only described the analysis of the Protein-Protein Interaction of key genes, but addressing important biological questions is left. Therefore, protein interaction experiments key genes must be provided.
Functional validation of protein-protein interactions falls beyond the scope of our meta-analysis work, as explained in the answer at point 2.
- In Discussion, the authors should discuss more on the differences of key drought tolerance-related genes between the previous studies and this study.
The Discussion has been completely rewritten, and several previous studies concerning the role of key drought tolerance-related genes have been discussed and compared to our study.
With regard to the four datasets used for the meta-analysis, we already considered to perform a comparison between the results of our meta-analysis and the results of the four selected papers. However, crucial differences were present between the used databases, tools and parameters of the previous papers. We just used the raw RNA reads available from the four selected papers, and we analysed them with specific pipelines that we specifically developed. In addition, the aims of these studies was quite different from ours, making difficult the comparison of the results. For this reason, it resulted very difficult to us to extract molecular data that could be compared with our data.
More in details:
Hordeum vulgare: in our analysis, RNA reads were mapped to IBSC v2 reference genome, while Cantalapiedra et al. (2017) de-novo assembled their reads and mapped the assembled transcripts to IBGSC reference, which is an older version than the one we used. The different assemblies leaded to different genes with different IDs (“HORVUxHrxGxxxxxx” for IBSC v2, “compxxxxx” for IBGSC), and this makes our results and Cantalapiedra et al. (2017) results not comparable.
Oryza sativa: The paper by Wei et al. (2017) is focused on alternative splicing, so their aim is completely different to the aim of our study. Indeed, the objects of the previous study are not the differentially expressed genes (DEGs) but the differentially expressed transcripts (DETs). This makes the results of the two studies not comparable. Anyway, we searched possible information about our 69 CDT genes in Wei et al. (2017), and we found three transcripts originated by three CDT genes in Supplementary Table 2. Nevertheless, we think that the comparison between the expression profile of one transcript and the expression profile of a gene generating additional transcripts would be not appropriate.
Brachypodium distachyon: Despite the use of the same reference genome, a proper table of DE analyses results is not reported by Lenk et al. (2019), making not possible a comparison between the two studies. The only key drought tolerance-related gene among our 69 CDT genes that was mentioned in the previous study is BRADI_3g11060v3 (Bradi3g11060) in Supplementary Table 5, but it is mentioned only as classified in GO:0055114 GO category (oxidation-reduction process), that is enriched in ABR8 (tolerant genotype) at time T2.
Zea mays: Despite the use of the same reference genome, the thresholds used for considering a gene as differentially expressed are substantially different between our study and the previous one. While we considered as DE a gene with FDR<0.05, Zenda et al. (2019) considered a FDR<0.001 and LFC>2, so the conditions they used were stricter than ours. Results of differential expression analyses of TC vs TD and SC vs SD are reported in Supplementary Files 5 and 6 of Zenda et al. (2019), respectively. Due to the different thresholds, only one of the 69 genes were found on file S5 and 10 genes were found on S6. For these genes, LFC values from the previous paper are really similar to the ones from our study.
Due to these differences, we think that the few results regarding a comparison between our study and those relative to the four papers used for the meta-analysis is not really interesting and would not give any added value to our study.
- The discussion section needs to be rewritten. In Discussion, I feel that the authors did not clarify the relationships between the genes and drought tolerance.
We strongly revised the Discussion, as highlighted in the PDF file Manuscript with the Word revision tracking. We integrated the new validation results and focused the discussion on the relationships between the identified genes and drought tolerance, in particular about the importance of the induction of leaf senescence and maintenance of photosynthesis during drought, and the transcription factors that may directly regulate the identified tolerance-related genes.
